# A Two-Stage Approach to the Study of Potato Disease Severity Classification

Yanlei Xu [1], Zhiyuan Gao [1], Jingli Wang [2], Yang Zhou [1], Jian Li [1] and Xianzhang Meng [2,*]

1    College of Information and Technology, Jilin Agricultural University, Changchun 130118, China; yanleixu@jlau.edu.cn (Y.X.); gzy@mails.jlau.edu.cn (Z.G.); zhouyang@jlau.edu.cn (Y.Z.); lijianperfect@163.com (J.L.)
2    College of Engineering and Technology, Jilin Agricultural University, Changchun 130118, China; wangjingli@jlau.edu.cn
*    Correspondence: jlnydx2009@163.com

**Abstract:** Early blight and late blight are two of the most prevalent and severe diseases affecting potato crops. Efficient and accurate grading of their severity is crucial for effective disease management. However, existing grading methods are limited to assessing the severity of each disease independently, often resulting in low recognition accuracy and slow grading processes. To address these challenges, this study proposes a novel two-stage approach for the rapid severity grading of both early blight and late blight in potato plants. In this research, two lightweight models were developed: Coformer and SegCoformer. In the initial stage, Coformer efficiently categorizes potato leaves into three classes: those afflicted by early blight, those afflicted by late blight, and healthy leaves. In the subsequent stage, SegCoformer accurately segments leaves, lesions, and backgrounds within the images obtained from the first stage. Furthermore, it assigns severity labels to the identified leaf lesions. To validate the accuracy and processing speed of the proposed methods, we conduct experimental comparisons. The experimental results indicate that Coformer achieves a classification accuracy as high as 97.86%, while SegCoformer achieves an mIoU of 88.50% for semantic segmentation. The combined accuracy of this method reaches 84%, outperforming the Sit + Unet_V accuracy by 1%. Notably, this approach achieves heightened accuracy while maintaining a faster processing speed, completing image processing in just 258.26 ms. This research methodology effectively enhances agricultural production efficiency.

**Keywords:** convolutional neural network; deep learning; disease classification; semantic segmentation; potato diseases; disease severity classification



## 1. Introduction

Potato holds the distinction of being the fourth largest staple food crop on a global scale and ranks as one of the most significant noncereal crops worldwide. Endowed with rich nutritional content, remarkable adaptability, and innate resistance, the potato assumes a pivotal role in agricultural production, economic progress, and human well-being [1–3]. However, within the growth trajectory of potato plants, the emergence of disease infestations on their leaves can precipitate a substantial decline in yield. Among these afflictions, early blight and late blight reign as the most pernicious and prevalent maladies befalling the potato crop [4]. The symptoms associated with both ailments manifest as scorching on tubers and leaves, presenting a formidable challenge in terms of differentiation and lending to widespread confusion [5]. Consequently, managing these diseases becomes an intricate task, underscoring the vital importance of early-stage detection and severity assessment. The evaluation of disease severity assumes particular significance as it paves the way for the adoption of targeted treatments and the implementation of adaptable pesticide dosing strategies, thus effectively curtailing control expenditures [6].

Generally, the severity of crop diseases is classified by dividing the degree of infection from mild to severe, represented by the percentage of area occupied by lesions at each

level [7–9]. Traditional assessment of potato disease severity relies mainly on the subjective judgment of experts, agricultural technicians, and farmers, which is time-consuming, labor-intensive, subjective, and inefficient, making it difficult to be widely applied [10]. With the rapid development of computer image processing, it has become extensively applicable to the task of grading the severity of crop diseases. Vision-based image processing segmentation techniques can automatically identify and segment spots by analyzing features such as patterns, shapes, colors, and textures in digital images, thus enabling disease severity classification [11]. Vision-based image processing segmentation techniques include threshold segmentation [12], region growth [13], and edge detection [14]. Chaudhary et al. [15] converted the RGB image into CIELAB, HIS, and YCbCr color space, respectively, based on different color features between the diseased spots and leaves, then based on the color features, the diseased spots were segmented by detecting the threshold value of diseased spots by applying OTSU method. Jothiaruna et al. [16] proposed a method combining color features and area growth for leaf lesion segmentation with an average segmentation accuracy of 87%. The above algorithms can achieve rapid segmentation of lesion areas; however, these algorithms are relatively simple. They yield limited segmentation results for complex backgrounds and diverse images with various changes. Additionally, they are sensitive to noise and image distortions. In recent years, with the development of artificial intelligence and deep learning, the technology of using deep learning to achieve disease severity classification has been proposed. Using image classification techniques to directly classify the severity of the pictures can provide fast severity assessment, and these algorithms for image classification include VGG [17], ResNet [18], MobileNet [19], ConvNeXt [20], and Transformer [21], etc. Chenghai Yin et al. [22] proposed DISE-Net to assess the severity level of maize leaflet spot disease, which is divided into five levels, and the recognition accuracy reaches 97.12%, but it can only achieve the grading of a single disease, and the requirements of the image are high. Meanwhile, the use of semantic segmentation techniques to segment leaf spots and thus grade the severity is also widely used, and these algorithms for semantic segmentation techniques include FCN [23], UNet [24], Deeplab [25], and PSPNet [26], etc. Hongbo Yuan et al. [27] proposed an improved DeepLab v3+ deep learning network for segmenting grape leaf black rot spots with an mIoU of 84.8%. Lin et al. [28] constructed a semantic segmentation model based on U-Net's convolutional neural network for segmenting cucumber powdery mildew spots with an mIoU of 72.11% and a Dice accuracy of 83.45%. Semantic segmentation techniques based on deep learning utilize deep neural networks to extract high-level semantic features, enabling more precise and fine-grained recognition and segmentation. While these segmentation algorithms can segment spots on plant leaves, traditional visual segmentation methods have high image quality requirements. On the other hand, deep learning-based image segmentation algorithms can effectively handle spots with complex backgrounds, displaying good generalization and robustness. In practical applications, employing deep learning-based semantic segmentation techniques to identify and segment spots on potato disease-infected leaves allows for rapid and accurate assessment of disease severity. Analyzing the results of lesion identification and segmentation enables efficient monitoring and prevention of the occurrence and spread of potato diseases. However, these methods can only achieve single-stage segmentation and severity level assessment for individual diseases, and their accuracy in assessment is relatively low.

A multi-stage severity assessment method has been proposed, which uses multiple deep-learning image processing tasks to process a single image for more accurate disease severity grading. These image-processing tasks encompass image classification, target detection, semantic segmentation, instance segmentation, and other techniques. Xueping Ni et al. [29] combined a target detection model and an image classification model to first detect blueberries using the MobileNet SSD target detection algorithm and then segmented the internal bruise and normal part using the MobileNet-UNet semantic segmentation model. The calculated percentage of the bruise, the average intersection and merger ratio (IoU) of blueberry segmentation and bruise segmentation were 0.979 and 0.773, respectively. While

this method is fast, it can only achieve the segmentation of a single case, and the evaluation accuracy is not high. Xudong Li et al. [30] combined an image classification model and a semantic segmentation model to effectively segment and detect potato leaf diseases amid complex backgrounds. They initiated the process by utilizing the Mask R-CNN instance segmentation technique to segment potato leaves within the image. Subsequently, they employed image classification models such as VGG16, ResNet50, and InceptionV3 to classify the diseases affecting potato leaves. Finally, they utilized semantic segmentation models including U-Net, PSPNet, and DeepLabV3+ to further classify potato leaves. The overall accuracy of this approach is commendable, yielding positive results. However, the three tasks are conducted independently of each other, which does not permit swift severity grading.

To enhance the precision and speed of disease severity assessment, an integrated approach that harnesses the strengths of diverse deep-learning tasks while optimizing the process for efficient and comprehensive grading is imperative. In this study, we introduce a two-stage methodology for classifying potato disease severity. This approach rapidly and accurately assesses the severity of early and late blight diseases in potatoes. Initially, we curated a dataset tailored to the two-stage task, serving both disease classification and semantic segmentation purposes. Subsequently, we devised a classification network named Coformer, primed to categorize potato images as exhibiting early blight, late blight, or being healthy. This model achieved a remarkable accuracy rate of 97.86% in disease classification. In the subsequent stage, we developed a semantic segmentation network drawing from Coformer and SegCoformer. This network effectively isolates diseased spots, leaves, and backgrounds within images, culminating in a segmentation mean Intersection over Union (mIoU) of 88.50%. Finally, we compute the ratio of diseased spots based on the segmentation outcomes and assess the severity of potato diseases in accordance with the national standard reference table for both early and late blight. Notably, this method achieves exceptional accuracy while maintaining an impressive processing speed of just 258.26 milliseconds per image. This study uniquely integrates the two-stage task of spot identification and spot segmentation, employing lightweight models for the severity grading endeavor. This innovative approach expedites the grading of disease severity across a diverse array of spots while ensuring the precision of severity assessment. Our methodology furnishes a dependable technical instrument for identifying and managing authentic potato diseases. The elevated precision and efficiency of this method empower swift and accurate classification of potato diseases, thereby contributing to the advancement of agricultural production efficiency. Furthermore, this study extends valuable insights to other classifications of crop disease, serving as a reference for future research endeavors.

## 2. Materials and Methods

### 2.1. Construction of the Dataset

In this study, we constructed two datasets with different tasks for the disease classification task and the semantic segmentation task of potato leaves with early and late blight. We obtained the data for the disease classification task by acquiring them from PlantVillage and Kaggle, and we collected a total of 3000 raw images, including healthy, early blight, and late blight leaves of potatoes. We resized the raw images to $256 \times 256$ pixels. We filtered the data for the semantic segmentation task by removing the images with angular deviation and incomplete leaves from the data for the disease classification task, and we obtained a total of 620 and 723 images of potato early and late blight, respectively. Then, we labeled the leaf and spot areas in the images using EISeg-1.0.2 software [31], with the normal areas of the leaves labeled in green, the spot areas in red, and the background labeled in black. Finally, we converted them into 8-bit depth maps for the semantic segmentation task.

### 2.2. Data Enhancement

We used convolutional neural networks for the training and validation of our model, and we required a large number of positive and negative samples. To improve the model's

generalization ability, robustness, and to prevent problems such as overfitting and sample bias, we performed data enhancement on the original dataset in this study [32]. In the first task stage of this study, we applied four data enhancement methods, as shown in Figure 1, including contrast enhancement (CE), brightness enhancement (BE), 20-degree rotation (RR), and color dithering (CD), which expanded the sample data to 5 times of the original one, with a total of 15,000 images. We also showed the sample maps before and after the image enhancement. For the classification task, we divided the dataset into training and validation sets according to a ratio of about 8:2 for the potato disease dataset, which were 12,000 and 3000 images, respectively.

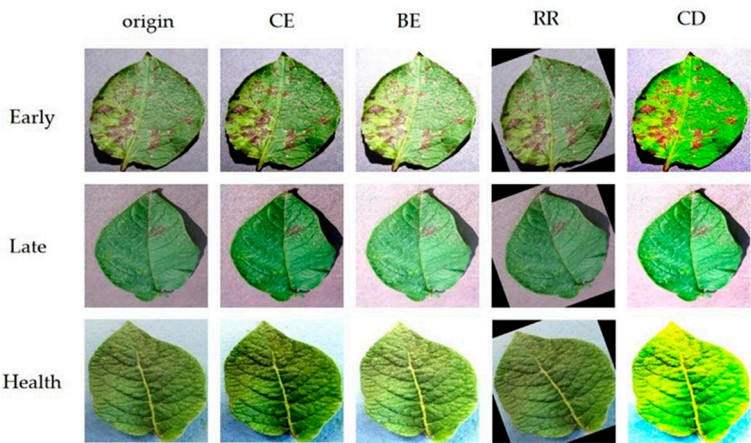

**Figure 1.** Data enhancement.

On the segmentation task, we synchronized the data enhancement of the labeled images and the corresponding labels of the images, respectively, and the enhancement method was horizontal flip, expanding the sample data to 2 times of the original, a total of 2686, and the dataset divided the potato disease dataset into the training, validation, and testing sets on the segmentation task according to the ratio of about 7:2:1. The specific data sample division is shown in Table 1.

**Table 1.** Division of data on the split task.

| Disease | Images | Training | Validation | Testing |
|---|---|---|---|---|
| early blight | 1240 | 868 | 248 | 124 |
| late blight | 1446 | 1013 | 289 | 144 |

### 2.3. Rules for Grading the Severity of Potato Diseases

Determining the severity of plant leaf disease is generally measured using the leaf spot coverage method, i.e., based on the ratio of the area covered by spots on the leaf to the area of the leaf. To estimate the severity of potato leaf disease, potato early blight leaves and late blight leaves were treated in this study, and the disease-infected spots on the leaves were separated from the normal part of the leaves, and the corresponding areas were calculated to compute the spot ratio. Then the severity of potato diseases was obtained according to the standard reference table of potato early blight [33] and late blight [34], as shown in Table 2, where Ke represents the standard reference of potato early blight and Kl represents the standard reference of potato late blight.

**Table 2.** Potato early blight and late blight standard reference table.

| Level | Severity | $K_e$/% | $K_l$/% |
|---|---|---|---|
| 0 | normalcy | $K_e = 0$ | $K_l = 0$ |
| 1 | relatively low | $0 < K_e \leq 5$ | $0 < K_l \leq 5$ |
| 3 | lower | $5 < K_e \leq 10$ | $5 < K_l \leq 10$ |
| 5 | medium | $10 < K_e \leq 20$ | $10 < K_l \leq 20$ |
| 7 | relatively high | $20 < K_e < 50$ | $20 < K_l < 50$ |
| 9 | high | $K_e \geq 50$ | $K_l \geq 50$ |

A semantic segmentation network segments the spot-covered part, the normal part, and the irrelevant background of an image, and outputs an image with only three labels: spot, normal, and background, and each pixel in the image is assigned a semantic label. In this study, the pixels of the three labels in the image were calculated and the spot ratio of potato diseases was obtained according to Equation (1). Assuming that the labeled image output from the semantic segmentation network is *M*, where each pixel takes the values of 0 (for background), 1 (for diseased spots), and 2 (for normal), the ratio of diseased pixels to the total number of pixels can be expressed according to Equation (1).

$$K = \frac{A_d}{A_d + A_l} \times 100\% = \frac{\sum\limits_{i,j}[M(i,j) = 1]}{\sum\limits_{i,j}[M(i,j) = 1] + \sum\limits_{i,j}[M(i,j) = 2]} \times 100\% \tag{1}$$

where $A_d$ is the area of diseased area, and $A_l$ is the area of the normal region, and $M(i,j) = 1$ denotes the counting of pixel points with a value equal to 1 in the *M* matrix, and $\sum ij$ denotes the summation of all pixel positions in the image $(i, j)$ in the image are summed.

*2.4. Experimental Flow*

The overall process of disease severity grading of potato early blight and late blight is shown in Figure 2. Firstly, the disease classification of potato leaves was carried out, which was divided into early blight, late blight, and healthy leaves. Those predicted by the model to be healthy leaves are directly outputted with a disease severity rating of 0. Images predicted by the model to be early blight or late blight are semantically segmented, with the leaf and spot portions segmented individually, and then the spot ratio Ke or Kl is computed, and finally, the disease severity rating is obtained by comparing with the reference table for early blight or early blight in potatoes.

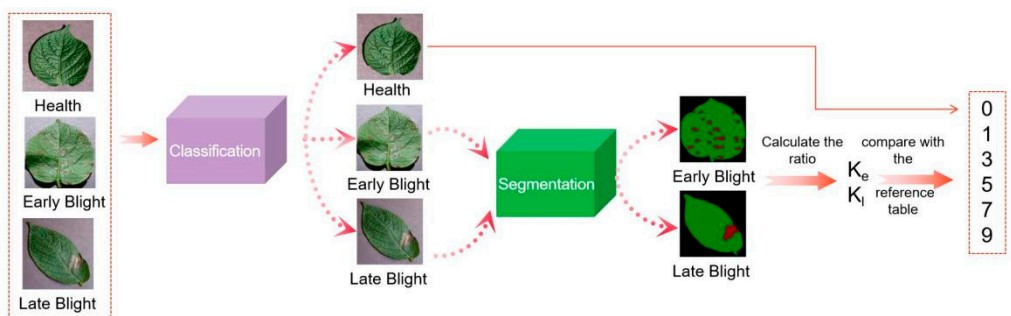

**Figure 2.** Experimental flow.

*2.5. Network Architecture*

2.5.1. Coformer

As shown in Figure 3, the Coformer network structure is combined by stacking ConvNeXt V2 block and Segformer block modules, which is the proposed classification network for potato diseases in this study. Firstly, the image is convolved to compress the size of the image feature map, and the number of channels is increased to enrich the feature map

information, then the key feature extraction of the main convolution module is performed, and finally pooling and normalization are performed for the prediction of disease classification. The ConvNeXt V2 block serves as the pivotal feature extraction module within the ConvNeXt V2 network [35]. This innovative block incorporates a depth-separable convolutional layer in tandem with a global response normalization layer, seamlessly operating as a unified convolutional process. Notably, this design empowers the network with exceptional generalization capabilities, all the while maintaining a parsimonious parameter count. Segformer block, as the core part of the Segformer network structure [36], i.e., the Transformer block in Figure 3, introduces a highly efficient self-attention mechanism, which accelerates and compresses the multiple self-attention mechanism to significantly reduces the amount of computation and improves the operation speed of the network. Simultaneously, the self-attention mechanism integrated into the Segformer block exhibits the capacity to dynamically ascertain correlations among individual positions. This adaptive learning capability significantly enhances the model's comprehension of semantic information within the image, thereby augmenting the precision of feature extraction processes.

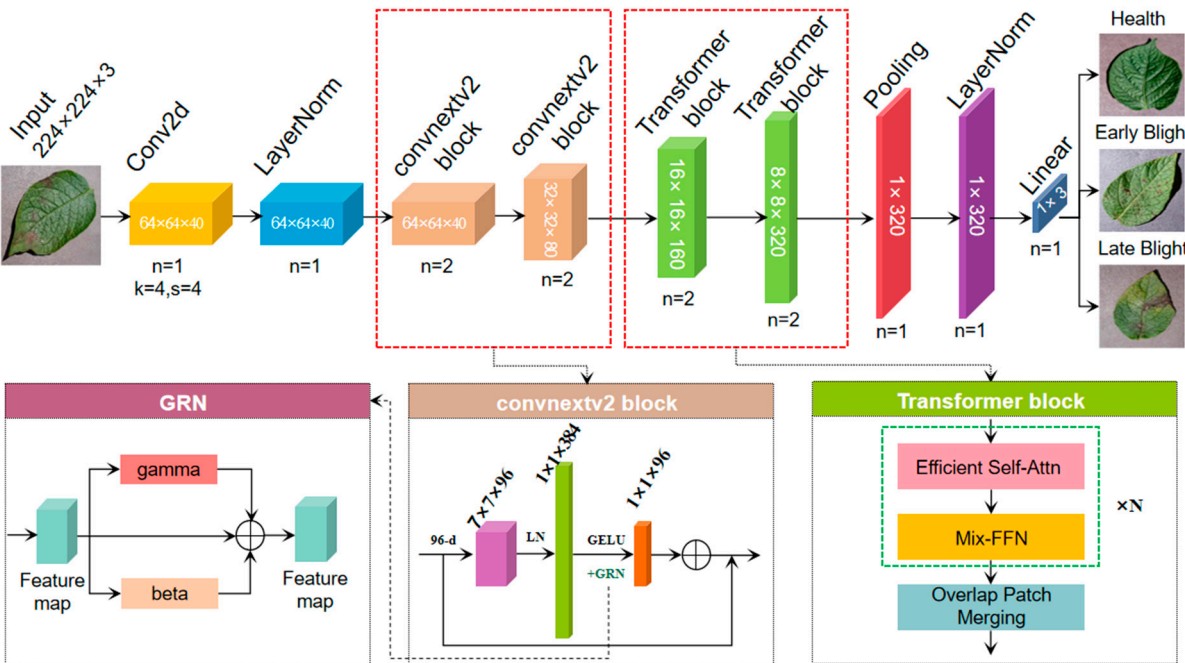

**Figure 3.** Structure of Coformer network.

In this study, the Coformer network structure is stacked in the combination of C-C-T-T [37], C denotes the convolutional neural network module and T denotes the Transformer module, which makes Coformer have the common advantage of the superb generalization ability of convolutional neural network and the whole area sensory field of the self-attention mechanism model. Additionally, in Section 3.1, we conduct a comparative analysis among three combination approaches: C-T-T-T, C-C-T-T, and C-C-C-T. The experimental findings consistently highlight the superior predictive accuracy of the C-C-T-T configuration. Furthermore, the Coformer has smaller model parameters and FLOPS, the first two convolutional modules use ConvNeXt V2 block for extracting deep features and improving the generalization ability of the model, and the last two modules use Segformer block to make the model get superb modeling ability. Here the ConvNeXt V2 block is referred to as 'block 1' and the Segformer block is referred to as 'block 2'. The Coformer network stack is shown in Table 3.

**Table 3.** Coformer network stacking.

| Operator | Input | Output | $n$ |
|---|---|---|---|
| Conv2d | $224 \times 224 \times 3$ | $64 \times 64 \times 40$ | 1 |
| LayerNorm | $64 \times 64 \times 40$ | $64 \times 64 \times 40$ | 1 |
| Convnext v2 block | $64 \times 64 \times 40$ | $64 \times 64 \times 40$ | 2 |
| Convnext v2 block | $64 \times 64 \times 40$ | $32 \times 32 \times 80$ | 2 |
| Transformer block | $32 \times 32 \times 80$ | $16 \times 16 \times 160$ | 2 |
| Transformer block | $16 \times 16 \times 160$ | $8 \times 8 \times 320$ | 2 |
| Pooling | $8 \times 8 \times 320$ | $1 \times 320$ | 1 |
| LayerNorm | $1 \times 320$ | $1 \times 320$ | 1 |
| Linear | $1 \times 320$ | $1 \times 3$ | 1 |

The integration of depth separable convolution within block 1 facilitates the extraction of profound features, enabling the network to emphasize specific regions in the image. This enhancement bolsters the model's perceptual performance, while the reduced parameter count of the depth separable network accelerates training and reasoning processes. Simultaneously, block 1 introduces Global Response Normalization (GRN) and applies it to the high dimensional features of each block 1. This action effectively removes redundant activations across channels, curbing the saturation of neurons and mitigating channel-related feature breakdowns. GRN within block 1 fosters significant biasing of neurons toward distinct patches of potatoes during feature extraction via the convolutional neural network. The spatial normalization of features amplifies the robustness and generalization of these features. Global response normalization operates as an adaptive normalization procedure on features, contingent on the feature distribution across the entire image. This operation decouples inter-feature dependencies, ultimately enhancing the model's expressive capacity. The GRN structure, as shown in Figure 3, encompasses a learnable scaling parameter 'gamma' for normalized feature scaling and a corresponding learnable bias parameter 'beta' for fine-tuning feature alignment across channels.

Block 2 contains the Efficient Self-Attention module of the transformer, which enables the whole network to obtain the global receptive fields, which are used to capture the contextual information at different locations in the input feature mapping so that the whole network obtains the deep feature information in the global context and performs the weighted aggregation of the information to obtain better classification results. Efficient Self-Attention calculates the attention by dividing the sequence into chunks and performing chunk matrix multiplication to accelerate the computation of self-attention and reduce the consumption of self-attention computation.

The computation of Efficient Self-Attention is shown in Equation (2). Firstly, three feature vectors are obtained from the input features, which are the data $Q$ (query), the $\overline{K}$ (Key) and $\overline{V}$ (Value). $\overline{K}$ and $\overline{V}$ are the feature embedding values mapped from high dimension to low dimension. Then the $Q$; and $\overline{K}$ perform a dot product operation and divide by $\sqrt{d}$ (where $d$ is the value of each $\overline{K}$ and $\overline{V}$ dimensions of the vectors) to obtain the score matrix for attention. Finally, the score matrix is compared with the $\overline{V}$ weighted summation and, by means of a learnable linear transformation (*softmax*), which again maps the feature map to the original input dimensions to obtain the final output of the multi-head attention mechanism.

$$Attention(Q, \overline{K}, \overline{V}) = softmax\left(\frac{Q\overline{K}^{T}}{\sqrt{d}}\overline{V}\right) \tag{2}$$

In addition, block 2 introduces Mix-FNN, which uses a $3 \times 3$ convolution in the feed-forward network (FFN) to convey the position information while considering the local information leakage caused by padding with zeros, which is able to avoid the problem of degradation of accuracy caused by interpolating position encoding (PE) at different resolutions. The calculation of Mix-FNN is shown in Equation (3). $x_{in}$ represents the

input features, i.e., the self-attentive output features, and $x_{\text{out}}$ represents the features of the output, and $MLP(x_{\text{in}})$ represents the pair of features that will be $x_{\text{in}}$ input multiple hidden layers for pass-through processing, and $GELU()$ represents the *GELU* activation function, compared to the RELU activation function adds stochastic regularity, which ensures efficient nonlinear output of the model and thus improves the robustness of the model, the $\text{Conv}_{3\times3}$ represents the $3 \times 3$ convolution operation.

$$x_{\text{out}} = MLP(GELU(Conv_{3\times3}(MLP(x_{\text{in}})))) + x_{\text{in}} \tag{3}$$

### 2.5.2. SegCoformer

While the current semantic segmentation model has achieved commendable outcomes in terms of segmentation accuracy, the intricate feature extraction and encoding/decoding procedures have resulted in a performance bottleneck concerning segmentation speed. In this research, an innovative approach is undertaken, building upon the foundation of the Coformer network. Specifically, the terminal Linear layer, referred to as the encoder, is replaced. Subsequently, a comprehensive Multi-Layer Perceptron (MLP) decoder is employed for the decoding process. This modified architecture, known as the SegCoformer network, is shown in Figure 4, depicting the structural configuration introduced in this study.

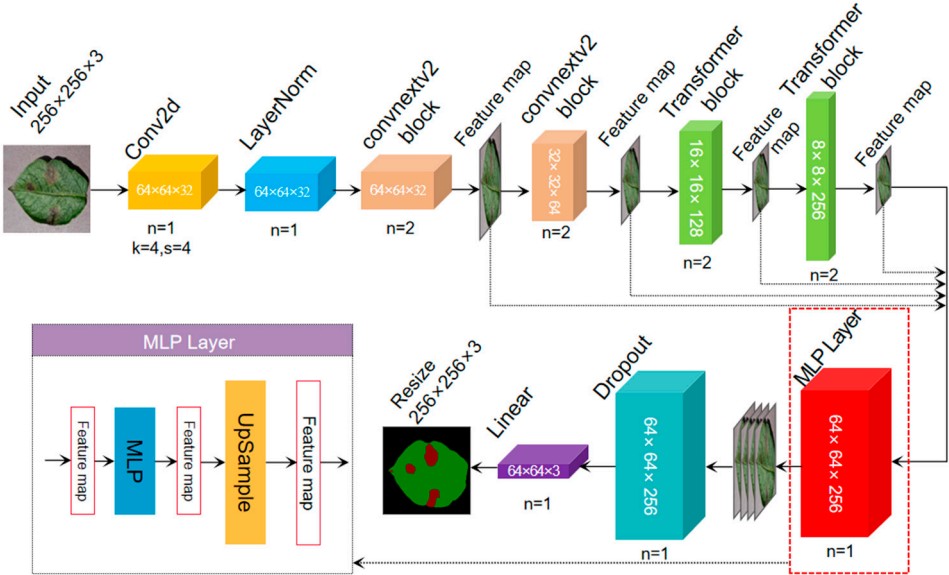

**Figure 4.** SegCoformer network structure diagram.

Specifically, the multilevel features produced by the encoder are initially harmonized across channels. This involves upsampling the feature map to a dimension equivalent to one-fourth of the input channels ($256 \times 256 \times 3$), as represented by Equation (4). The 1/4 times feature map is then passed through the MLP layer to fuse the cascaded features F^ and the MLP layer is used to project the fused features F^ into the segmentation mask M with $H/4 \times W/4 \times N_{\text{cls}}$ resolution, and the M is computed as shown in Equation (5), where $N_{\text{cls}}$ is the number of categories. Finally, the feature map with $H/4 \times W/4 \times N_{\text{cls}}$ size is reduced back to the size of $H \times W \times N_{\text{cls}}$, i.e., ($256 \times 256 \times 3$). Compared to traditional decoders such as ASPP [38], this module receives the depth feature map from the encoder as input and feeds it into multiple null convolution branches with different expansion rates, multiple sets of null convolutions are able to obtain different sizes of the sensory fields, followed by global average pooling of the feature maps, and finally, splicing together the features of the three branches and upsampling them to the same size as the input feature map. The MLP decoder can effectively reduce the number of network parameters and computation amount while obtaining the global receptive field compared with the traditional ASPP module, which can fuse cross-channel feature information and obtain

different sizes of receptive fields, but it will increase the network complexity, the number of parameters, and the computation amount accordingly. In contrast, the MLP decoder is a more lightweight solution, which is suitable for converting the depth feature maps output from the encoder into high-resolution prediction results.

$$\hat{F}_i = \text{Upsample}\left(\frac{H}{4} \times \frac{W}{4}\right)\left(\hat{F}_i\right), \forall I \tag{4}$$

$$M = \text{Linear}(C, N_{cls})\left(\text{Linear}(4C, C)\left(\text{Concat}(\hat{F}_i)\right), \forall i\right) \tag{5}$$

where Upsample (H × W) (−) denotes upsampling the feature map to H × W dimensions, Linear ($C_{in}$, $C_{out}$) (−) denotes the linear layer with $C_{in}$ and $C_{out}$ as input and output vector dimensions, respectively, and M denotes the mask values of the different categories predicted by the model. The SegCoformer network stack is shown in Table 4.

**Table 4.** SegCoformer network stacking.

| Operator | Input | Output | *n* |
|----------|-------|--------|-----|
| Conv2d (k = 4, s = 4) | 256 × 256 × 3 | 64 × 64 × 32 | 1 |
| LayerNorm | 64 × 64 × 32 | 64 × 64 × 32 | 1 |
| Convnext v2 block | 64 × 64 × 32 | 64 × 64 × 32 | 2 |
| Convnext v2 block | 64 × 64 × 32 | 32 × 32 × 64 | 2 |
| Transformer block | 32 × 32 × 64 | 16 × 16 × 128 | 2 |
| Transformer block | 16 × 16 × 128 | 8 × 8 × 256 | 2 |
| MLP Layer | 8 × 8 × 256 | 64 × 64 × 256 | 1 |
| Dropout | 64 × 64 × 256 | 64 × 64 × 256 | 1 |
| Linear | 64 × 64 × 256 | 64 × 64 × 3 | 1 |
| Resize | 64 × 64 × 3 | 256 × 256 × 3 | 1 |

*2.6. Experimental Environment*

In order to ensure the fairness and reproducibility of the experimental results, Windows 10 was chosen for this study in terms of the operating system. Specifically, the CPU model of the computer was Intel Core (TM) i7-7820X with 2.4 GHz and 32 GB of RAM. Meanwhile, the graphics processor model was NVIDIA TITAN Xp, with a graphics memory of 12 GB. The software environment used in the experiment is Python 3.8 and Pytorch-GPU 1.9.

*2.7. Experimental Parameters*

In order to make the experimental results more comparable and at the same time to make an objective comparison of the performance of different models, this study sets uniform experimental parameters. In the image classification task, firstly in the normalization of each image of the input, Equation (6) "output" denotes the output after normalization and "input" denotes the input image, and "mean" denotes the mean value, which is taken as (0.485, 0.456, 0.406), "std" denotes the standard deviation, which is taken as (0.229, 0.224, 0.225). The optimizer uses AdamW (Adaptive momentum and weight decay), the loss function uses the cross-entropy loss function, and the training uses the migration learning technique in order to improve the model's generalization [39], pretraining weights of different networks on imagenet-1k are used separately (except MobileVit and Coformer) [40], the batch size is set to 64, the learning rate is initialized to 0.001, and the number of iterations is 50.

$$\text{output} = \frac{\text{input} - \text{mean(input)}}{\text{std(input)}} \tag{6}$$

In the semantic segmentation task, in order to reduce overfitting, the training strategy of migration learning (except SegCoformer) is used, where the weights are frozen for the initial 100 rounds and unfrozen for the next 100 rounds for fine-tuning. The pre-training weights of the different networks on imagenet-21k are used separately, the optimizer is

used with AdamW, and the loss function is used with the cross-entropy loss function. The learning rate is initialized to 0.001, the weights decay to 0.01, the batch size is set to 16 and the total number of iterations is 200.

*2.8. Indicators for Model Evaluation*

Considering that the image classification task and the semantic segmentation task are different computer vision tasks, precision (%), recall (%), F1 score (%), and accuracy (%) were chosen as the metrics to evaluate the performance of the classification model for better testing the performance of the proposed model in this study.

$$\text{Precision} = \frac{\text{TP}}{\text{TP} + \text{FP}} \tag{7}$$

$$\text{Recall} = \frac{\text{TP}}{\text{TP} + \text{FN}} \tag{8}$$

$$\text{F1}\_\text{score} = \frac{2\text{TP}}{2\text{TP} + \text{FP} + \text{FN}} \tag{9}$$

$$\text{Accuracy} = \frac{\text{TP} + \text{TN}}{\text{TP} + \text{FP} + \text{FN} + \text{TN}} \tag{10}$$

where TP, TN, FP, and FN are the number of true positive samples, true negative samples, false positive samples, and false negative samples, respectively. Precision is a measure of the proportion of samples that the model correctly predicts as positive given all samples predicted as positive, as in Equation (7). Recall measures the proportion of all true positive samples in which the model successfully finds a positive, as in Equation (8). The F1-score is the reconciled average of precision and recall, and is used to combine the accuracy and recall of the classifier, as in Equation (9). Accuracy is the most intuitive measure of model quality, and it describes the proportion of all samples correctly classified by the classifier, as in Equation (10). In this study, precision, recall, Dice coefficient, and mean intersection and merger ratio (mIoU, %) are chosen as metrics for evaluating the performance of semantic segmentation models.

The Dice coefficient, as in Equation (11), is usually used to calculate the similarity of two samples, and its value ranges from (0, 1). The Dice coefficient is defined as the ratio of the twice of the intersection between the prediction mask and the true label mask to the sum of the number of pixels of the prediction mask and the true label mask. In situations where the prediction is in perfect agreement with the ground truth, the Dice coefficient reaches its maximum value of 1. Conversely, if the prediction and ground truth do not align at all, the Dice coefficient attains a minimum value of 0. Therefore, the higher the Dice coefficient, the closer the model prediction results are to the true labels, and the better the model performance.

$$\text{Dice} = \frac{2\text{TP}}{\text{FP} + 2\text{TP} + \text{FN}} \tag{11}$$

mIoU, as in Equation (12), as a metric for model performance evaluation, can reflect the model's table on semantic segmentation tasks more objectively. We used mIoU as the main metric to measure the performance of our method. Additionally, we calculated the IoU value of each category for each image in the test set and weighted the average to obtain the mIoU of the whole image.

$$\text{mIoU} = \frac{1}{k+1}\sum_{i=0}^{k} \frac{P_{ii}}{\sum_{j=0}^{k} P_{ij} + \sum_{j=0}^{k} P_{ji} - P_{ii}} \tag{12}$$

where, $P_{ij}$ represents the number of pixels that originally belong to class i but are predicted to be class j, $P_{ii}$ class i, and $P_{ji}$ represents the number of pixels that are truly labeled as class i and are predicted to represent the number of pixels that originally belonged to class j but are predicted to be class i. In this study, the pixels in each image are classified into three

classes: speckle, blade, and background, so k is taken as 2. To better measure the complexity of the model, this study uses the number of parameters (params) and the amount of floating point calculations (FLOPs). The FLOPs are calculated as shown in Equation (13).

$$\text{FLOPs} = \left[\sum_{\text{Conv}=n}^{n=1} \left(2C_iK^2 - 1\right) HWC_O\right] + \left[\sum_{\text{Full}=n}^{n=1} (2I - 1)O\right] \tag{13}$$

where $C_i$ is the number of input channels of the ith convolutional layer, $C_O$ is the number of output channels of the convolutional layer, K is the convolution kernel size, and H is the height and width of the output feature map of the convolutional layer, respectively. W are the height and width of the output feature map of the convolutional layer, respectively, I is the height and width of the output feature map of the convolutional layer, and O is the number of inputs and outputs in the fully connected layer, respectively.

## 3. Tests and Analysis of Results

### 3.1. Coformer Network Classification Performance Analysis

To validate the various Coformer combinations outlined in Section 2.5.1, training experiments were performed on the image classification dataset utilized in this study. The networks encompassing the combinations C-C-C-T, C-T-T-T, and C-C-T-T-T were individually trained and designated as Coformer1, Coformer2, and Coformer. The comparative outcomes are detailed in Table 5.

**Table 5.** Comparison results of different combinations of Coformer.

| Network | Accuracy/% | F1-Score/% | Params/M | Flops/G |
|---|---|---|---|---|
| Coformer1 | 95.56 | 95.56 | 3.58 | 0.43 |
| Coformer2 | 96.33 | 96.34 | 4.46 | 0.52 |
| Coformer | 97.86 | 97.87 | 4.06 | 0.46 |

Analysis of Table 3 shows that Coformer1 has the worst performance with only 95.56% accuracy but has the smallest model complexity among the three networks. Coformer2 outperforms Coformer1 with 96.33% accuracy, which is 0.77% higher compared to Coformer1 but has the largest amount of model parameters and FLOPs. Coformer, on the other hand, balances the model complexity while ensuring the highest model performance with an accuracy of 97.86%, which is 1.53% higher than Coformer2. For further analysis, their confusion matrices are plotted separately, as shown in Figure 5.

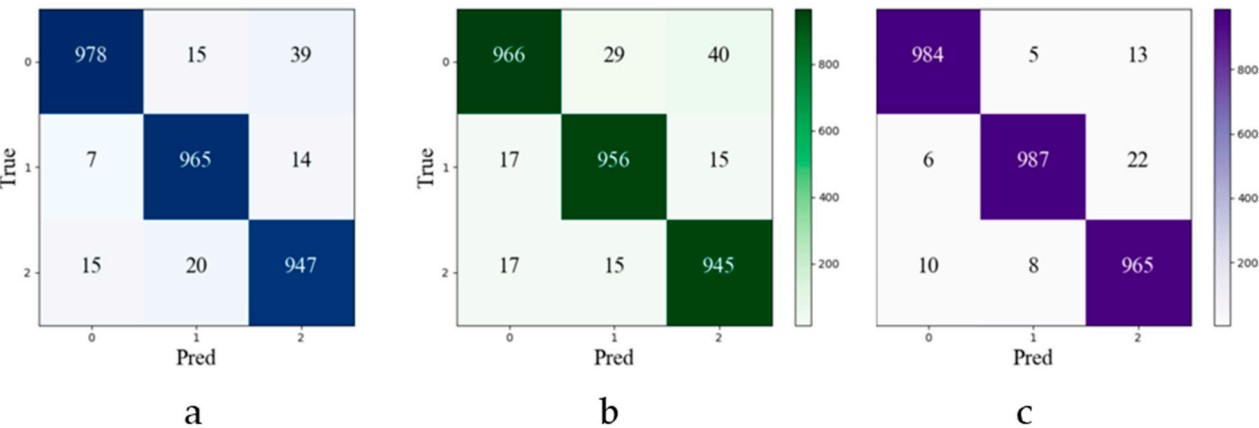

**Figure 5.** Confusion matrix ((**a**) represents the confusion matrix for Coformer1, (**b**) represents the confusion matrix for Coformer2, (**c**) represents the confusion matrix for Coformer, True represents the true value, and Pred represents the model predicted value).

The analysis reveals that all three models are generally proficient in making accurate predictions, albeit with minor differences. Coformer exhibits a higher count of correctly predicted samples compared to Coformer1 and Coformer2. Through comprehensive analysis, Coformer amalgamates the advantages of CNN models and Transformers into a singular architecture, yielding the best performance. It excels in robust generalization and demonstrates commendable modeling capabilities simultaneously.

### 3.2. Comparison of Classification Performance of Coformer Network Compared to Other Transformer Networks

To validate the superiority of the Coformer network over other feature extraction networks containing Transformers, experimental comparisons are conducted with alternative Transformer based networks, which are ViT [41] (vision_transformer_base_patch16_224), Sit (swin_transformer_tiny_patch4_window7_224) [42], CoAt (CoAtNet0) [37], MVit (mobile_vit_small) [19], and Coformer, which are the versions with the smallest number of parameters, are analyzed for performance and model complexity. The results of the comparison are shown in Table 6.

**Table 6.** Comparison results with other Transformer models.

| Network | Accuracy/% | Precision/% | Recall/% | F1-Score/% | Params/M | Flops/G | Size/M |
|---------|-----------|-------------|----------|------------|----------|---------|--------|
| Vit | 98.35 | 98.36 | 98.36 | 98.36 | 85.64 | 16.86 | 327.35 |
| Sit | 99.06 | 98.60 | 99.03 | 98.96 | 98.81 | 15.16 | 105.27 |
| CoAt | 98.06 | 98.07 | 98.07 | 98.07 | 16.99 | 3.35 | 66.57 |
| MVit | 93.89 | 92.02 | 94.70 | 93.34 | 4.94 | 1.46 | 19.00 |
| Coformer | 97.86 | 97.87 | 97.87 | 97.87 | 4.06 | 0.46 | 10.19 |

Analyzing Table 6 shows that among the five networks under comparison, Coformer boasts the least intricate model structure and showcases commendable performance. While Vit exhibits the highest FLOPs and model size, Sit claims the largest parameter count. However, Sit's superiority in performance can be attributed to its layered local attention mechanism. As for CoAT, its model complexity falls within the intermediate spectrum; even though its parameter count, FLOPs, and model size are reduced in comparison to Vit and Sit, its accuracy lags behind Vit's. MVit adopts separable convolution and attention mechanisms to alleviate the model's computational demands and parameter volume. Although MVit significantly streamlines network complexity in contrast to Vit, Sit, and CoAt, the reduction in parameters leads to a notable performance decline. It ranks as the poorest performer among all networks, with accuracy and precision rates of 93.89% and 92.02%, respectively. This suggests that the model lacks high confidence in classifying samples as positive instances. Additionally, MVit exhibits the lowest recall rate, standing at only 94.70%, implying that the model frequently misclassifies true positive samples as negative. On the other hand, in comparison, Coformer touts the least intricate network complexity while still approaching CoAT's performance, with a marginal accuracy difference of just 0.2%. This underscores Coformer's ability to maintain minimal model complexity while achieving respectable performance. As illustrated in Figure 6, for a more comprehensive analysis of Coformer's performance, this study plots training loss, training accuracy, validation loss, and validation accuracy across various epochs.

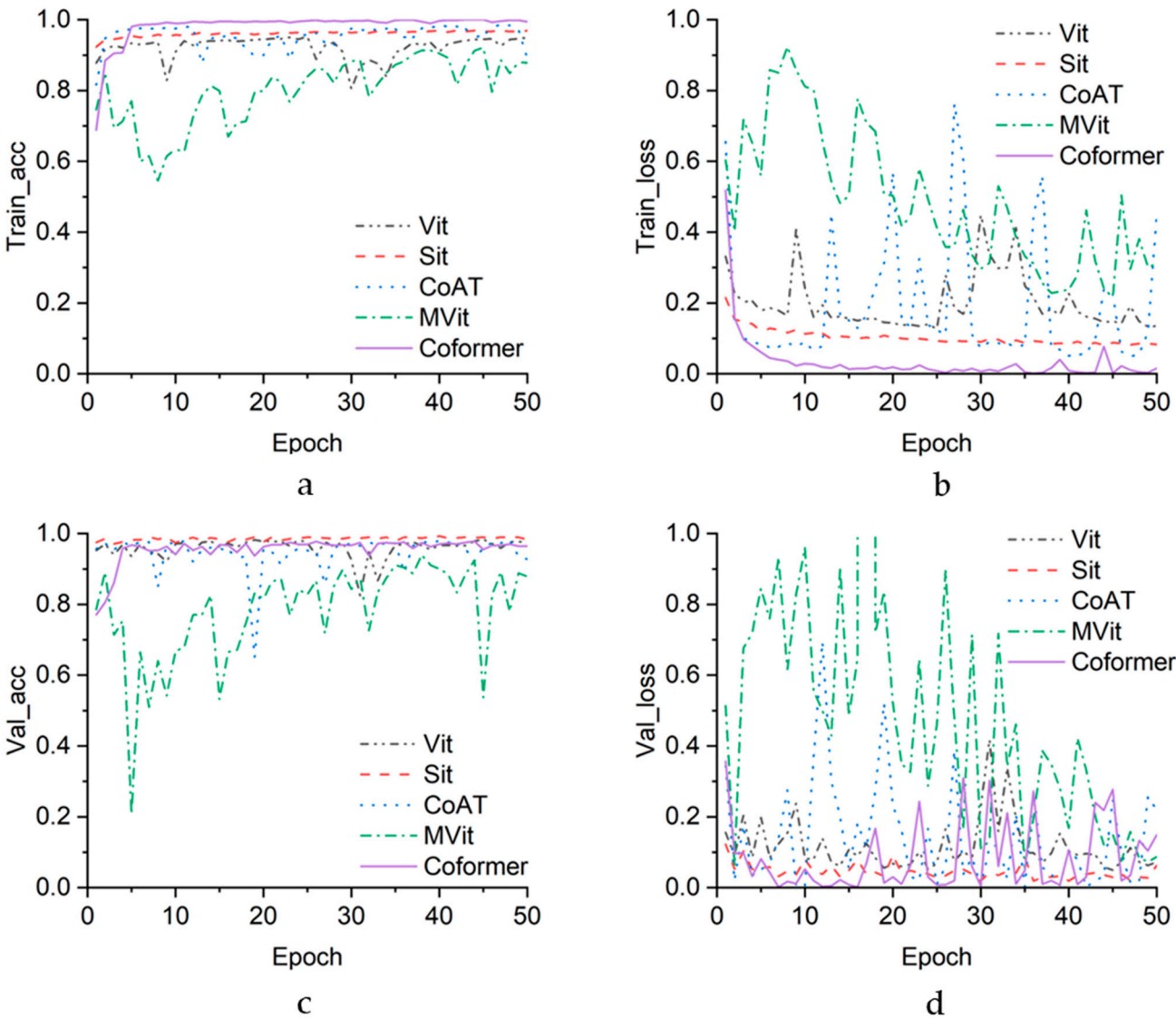

**Figure 6.** Comparison of training process ((**a**) represents training set accuracy, (**b**) represents validation set accuracy, (**c**) represents training set loss, and (**d**) represents validation set loss).

From Figure 6, it becomes apparent that throughout the training process, all networks except for the MVit and Coformer networks are equipped with corresponding pretraining weights. As a result, the entire training process displays greater stability, characterized by minor fluctuations, and achieves a high level of accuracy within the initial five rounds. Conversely, the MVit and Coformer networks, lacking pretraining weights and featuring lower network complexity, exhibit more substantial fluctuations in both training and validation during the initial five rounds. Nevertheless, across the entirety of the training process, Coformer outperforms MVit in terms of fluctuation in loss values, variation in accuracy rates, and the magnitude of accuracy. Among the networks, Sit demonstrates the highest stability throughout the training process, marked by minimal fluctuation amplitude. Simultaneously, it attains the highest validation accuracy. While Coformer does not maintain peak performance throughout the entire process, it still surpasses CoAT and MVit, without demonstrating signs of overfitting. Coformer has the strongest overall performance in terms of accuracy and speed in disease classification, and it can provide efficient and accurate support for the rapid grading of potato disease severity.

### 3.3. Performance Analysis of Semantic Segmentation of SegCoformer Model

In order to evaluate the training effect and generalization ability of the SegCoformer model, this study plots the overall training process of the SegCoformer model, as shown in Figure 7, where the main axis shows the trend and change of the mIoU values of the model on the validation set in each round, and the subaxis shows the trend and change of the training loss and validation loss. Analyzing Figure 7 shows that during the initial stages of training, both the training loss and validation loss are elevated, accompanied by low mIoU values. However, as the training progresses, there is a consistent decline in both the training and validation losses, while the mIoU values demonstrate a steady ascent. Around the 180th round of training, the model attains an mIoU value of 88.50%, indicating its proficient capability in accurately segmenting images with a high degree of precision. Notably, by the 185th round, the model achieves its lowest training and validation losses. On the whole, these findings underscore a highly favorable outcome, signifying the successful acquisition of segmentation capabilities with remarkable accuracy by the SegCoformer model.

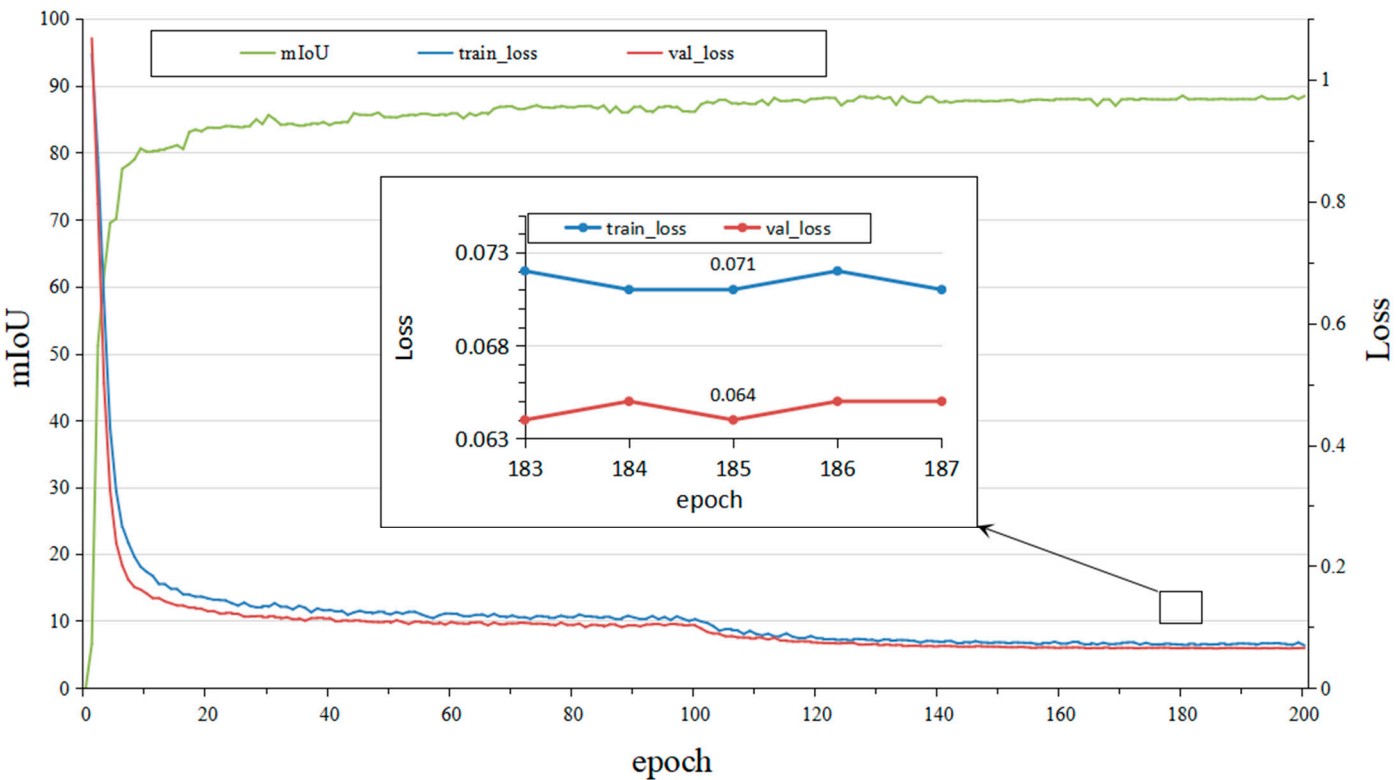

**Figure 7.** Diagram of the training process.

We used the model with the highest mIoU value to segment an image of early blight on a potato leaf and another image of late blight. The segmentation results are depicted in Figure 8. In this figure, "origin" denotes the original image, "a" represents the image after background removal-essentially the segmented target image, "b" showcases the segmented image, where green corresponds to the healthy parts of the leaf, red indicates the diseased portions, and "c" displays a combination of the original image and the segmented image, with a mixing factor of 0.7. From the figures, it is evident that the SegCoformer model delivers exceptional accuracy in segmenting the primary leaf structure. This holds particularly true for the early blight image, where numerous shadows were predicted correctly. Furthermore, the SegCoformer model exhibits remarkable precision in segmenting the disease spots, capturing even the minutest details of both early and late blight spots.

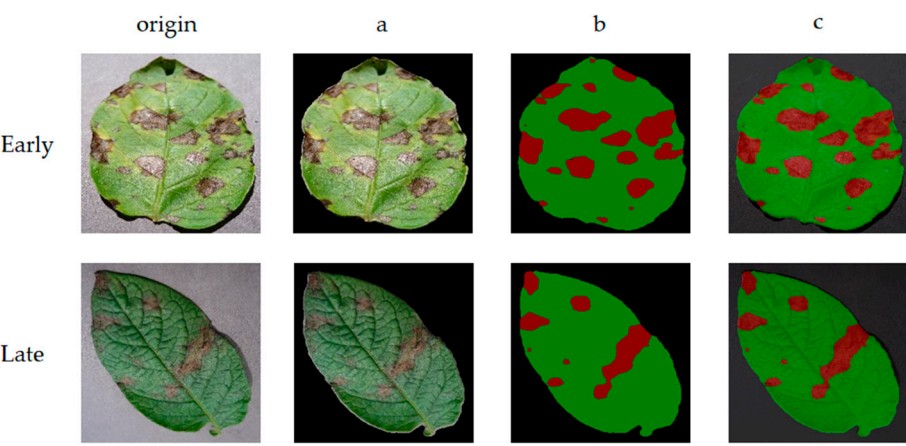

**Figure 8.** Segmentation result graph.

*3.4. Performance Comparison of SegCoformer Compared to Other Semantic Segmentation Network Parameters*

In order to verify the segmentation performance of SegCoformer, based on the dataset of the semantic segmentation task of this study, it was compared with other semantic segmentation networks Deep_X (deeplabv3_xception) [43], Deep_M (deeplabv3 + mobilenetv2), Unet_V (u-net_vgg16), Unet_R (u-net_resnet50), Segformer (Segformer_b0) [36], and SegCoformer were experimentally compared to analyze their performance and model complexity for segmenting spots of early and late blight in potato. According to the analysis in Table 7, it can be seen that in terms of mIoU and Dice, Unet_V is the most effective, with mIoU reaching 89.85% and Dice reaching 94.42%, so the Unet_V model is the most effective in segmenting the whole image. In terms of network complexity, Segformer is the smallest, with only 3.71 M parameters and 3.38 G FLOPs; in terms of model size, also Segformer is the smallest, with only 14.24 MB, but the segmentation effect of Segformer is not too good. Additionally, the network complexity difference between SegCoformer and Segformer is very small, the mIoU of SegCoformer reaches 88.50%, which is 1.91% higher than that of Segformer. Moreover, compared to Unet_V, the mIoU value is 1.35% lower, but the number of parameters, floating-point computation, and model size of SegCoformer is 4.51 M, 3.57 G, and 17.5 MB, which are about 11 times, 12 times and 12 times less compared to Unet_V, respectively. SegCoformer has strong segmentation performance and small computation consumption, which makes it a comprehensive and powerful semantic segmentation network.

**Table 7.** Performance comparison with other semantic segmentation network parameters.

| Model | mIoU/% | Dice/% | Precision/% | Recall/% | Param/M | GFLOPS/G | Size/MB |
|---|---|---|---|---|---|---|---|
| Deep_X | 86.47 | 92.42 | 93.54 | 91.33 | 54.70 | 41.71 | 209.70 |
| Deep_M | 88.28 | 93.47 | 94.24 | 92.72 | 5.81 | 13.21 | 22.40 |
| Unet_V | 89.85 | 94.42 | 94.99 | 93.86 | 24.89 | 112.92 | 94.97 |
| Unet_R | 89.55 | 93.08 | 94.95 | 91.29 | 43.93 | 46.03 | 167.90 |
| Segformer | 86.59 | 92.01 | 92.76 | 91.28 | 3.71 | 3.38 | 14.24 |
| SegCoformer | 88.50 | 93.90 | 95.02 | 92.81 | 4.51 | 3.57 | 17.51 |

Different models exhibit varying segmentation performance for the two types of diseases, which we plotted as a double bar graph in order to evaluate the models more intuitively, as shown in Figure 9. The optimal performance in early disease segmentation was achieved by Unet_V, attaining an IoU value of 88.73%. For late disease segmentation, SegCoformer proved to be the most effective, yielding an IoU value of 91.57%. When considering prediction precision across various categories, SegCoformer stands out as the

top performer. The best recall is also SegCoformer with 92.81%, which shows that the model has good generalization ability and stability.

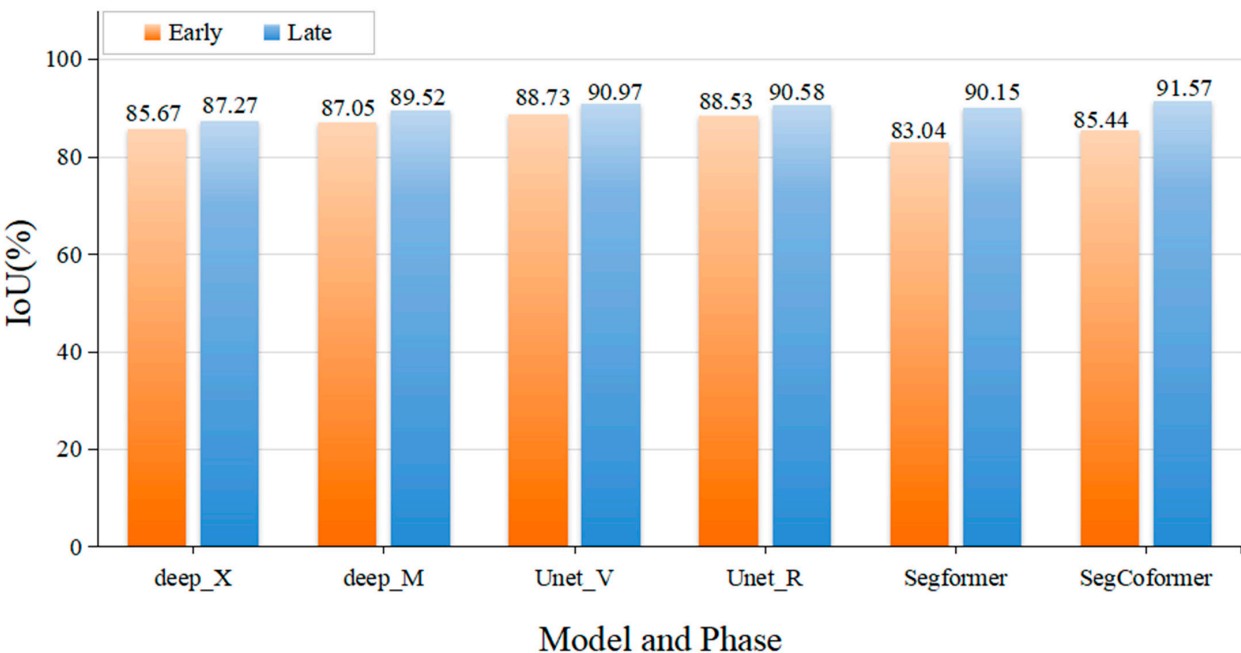

**Figure 9.** IoU of different models segmenting two diseases.

Specifically, for the actual segmentation effect of the model, we selected deep_M, Unet_V, Segformer, and SegCoformer, which have the highest mIoU, for the actual segmentation comparison experiment, as shown in Figure 10. Analysis shows that, for potato early blight segmentation results, Unet_V showcases the most superior segmentation performance, excelling in capturing even the minutest spots with remarkable detail. Conversely, Segformer's segmentation performance is relatively subpar, as it struggles to identify numerous small spots. SegCoformer and deep_M demonstrate similar performances, not reaching the level of Unet_V, a distinction evident in Unet_V's highest IoU score. Switching to potato late blight segmentation results, SegCoformer notably emerges as the leader among the various networks, boasting the most robust segmentation outcomes, complete with intricate details. Comprehensively, SegCoformer is a semantic segmentation network with strong segmentation performance and at the same time possessing smaller network complexity.

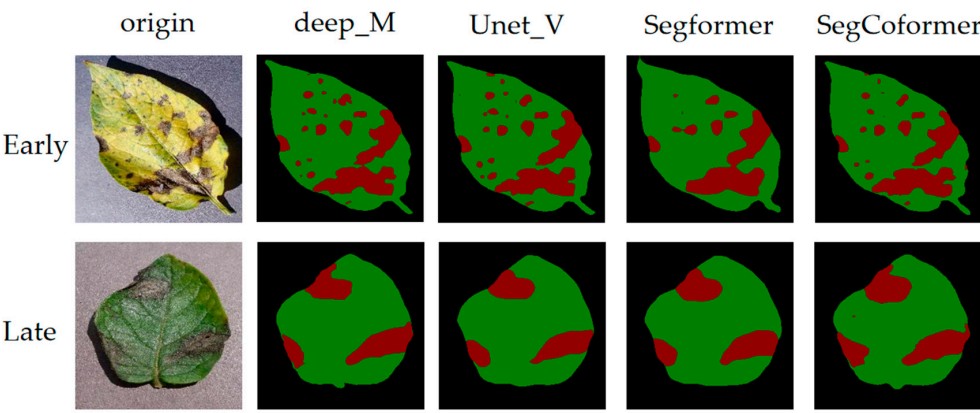

**Figure 10.** Comparison of segmentation for different models.

### 3.5. Comparative Experiment on Potato Disease Severity Grading

To assess the effectiveness of the Coformer and SegCoformer models in the task of grading the severity of actual potato diseases, this study employed a set of 100 potato leaf images from the test dataset. These images were used as samples for conducting tests related to image classification, image segmentation, and severity grading. Among these samples, there were 40 images of early blight, 40 images of late blight, and 20 images of healthy potato leaves. The Coformer model was utilized for the classification task to predict the disease category for each potato leaf. The SegCoformer model was employed for the semantic segmentation task to obtain disease segmentation outcomes for each potato leaf. By utilizing the severity grading rules proposed in this study, a severity grading value (K value) was assigned to each test sample. Subsequently, a comparison was performed between the predicted outcomes and the ground truth labels for these samples. Sit has the highest accuracy on the classification task and Unet_V has the highest mIoU value on the classification task, so the test comparison was conducted with Sit as the object of the classification algorithm and Unet_V as the object of the segmentation algorithm, respectively, as shown in Table 8, which shows the test results of the first 10 samples only.

**Table 8.** Test results for the first 10 samples.

| No. | True Value | | | Coformer | | | SegCoformer | | Sit | | Unet_V | |
|---|---|---|---|---|---|---|---|---|---|---|---|---|
| | Class | K/% | Grade | Class | K/% | Grade | | | Class | K/% | Grade |
| 1 | Health | 0 | 0 | Health | 0 | 0 | Health | 0 | 0 | | | |
| 2 | Late | 5 | 1 | Late | 5 | 1 | Early | 6 | 3 | | | |
| 3 | Late | 9 | 3 | Late | 11 | 5 | Late | 11 | 5 | | | |
| 4 | Early | 9 | 3 | Early | 10 | 3 | Early | 10 | 3 | | | |
| 5 | Early | 16 | 5 | Early | 15 | 5 | Early | 17 | 5 | | | |
| 6 | Late | 17 | 5 | Late | 18 | 5 | Late | 11 | 5 | | | |
| 7 | Health | 0 | 0 | Health | 0 | 0 | Health | 0 | 0 | | | |
| 8 | Health | 0 | 0 | Health | 0 | 0 | Health | 0 | 0 | | | |
| 9 | Late | 39 | 7 | Late | 45 | 7 | Late | 49 | 7 | | | |
| 10 | Early | 9 | 3 | Early | 9 | 3 | Early | 10 | 3 | | | |

Analyzing Table 8, it can be learned that on the classification task, the Coformer and Sit proposed in this study are accurate for potato early blight, late blight and healthy leaf images. On the segmentation task, compared to the ground truth values, SegCoformer exhibits an average error of 2.04% in predicting K values, while Unet_V demonstrates an average absolute error of 1.46% in predicting K values. Unet_V has a slightly smaller average error, and the K-value results for both models indicate that the differences between the actual and predicted outcomes are not significant.

SegCoformer achieved a testing accuracy of 83% in assessing the severity of these 100 samples, while Unet_V achieved an accuracy of 84%. Both models demonstrated relatively high actual precision in predicting the severity of disease on individual leaves. To provide a comprehensive comparison, combining the insights from this table and other experiments, we created a contrast between the Coformer + SegCoformer combination and the Sit + Unet_V combination for grading the severity of potato diseases. We visually presented this multidimensional data comparison through a corresponding bar graph, as shown in Figure 11. The relevant parameters encompass classification performance (CP/%), segmentation performance (SP/%), the average error in K value prediction on the test set (KE/%), testing accuracy on the test set (TSA/%), comprehensive time for a single image (CT/ms), and total floating-point computation (TFO/G). Due to the different weights of the samples, we normalized these parameters in order to compare multiple parameters from different models. For positive weights, we set the maximum value to 1 and divide the other values by the maximum value to obtain the ratio of the parameters. For example, if the maximum value of parameter A is 10 and the minimum value is 2, the normalized parameter A is a scaled value between 0.2 and 1. Conversely, for inverse weights use

inverse normalization. In this way, we can eliminate the effect of parameter units and better compare the relative sizes of different parameters.

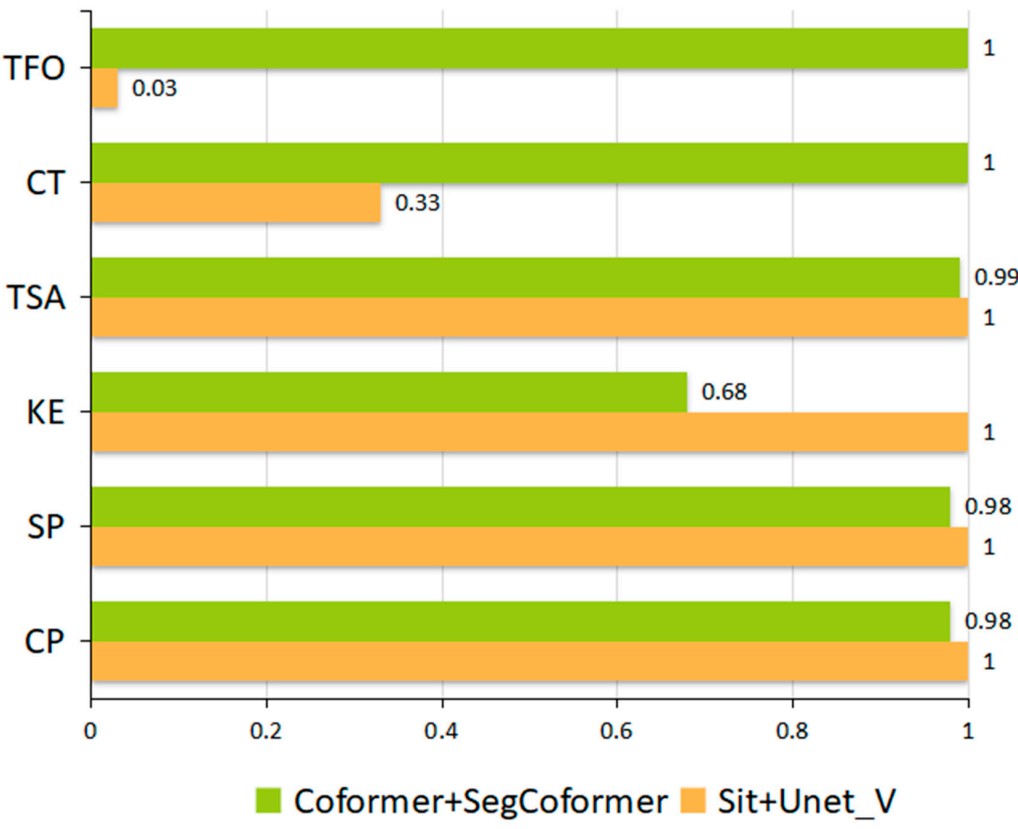

**Figure 11.** Comparison of the combined performance of the two combinations.

Analyzing Table 9 and Bar Graph 11 shows that the performance of Coformer + SegCoformer is slightly lower than Sit + Unet_V, and the correctness of the test set is almost equal. The inference speed of Coformer + SegCoformer is significantly better than that of Sit + Unet_V, and in terms of the total floating-point computation, Coformer+ SegCoformer is absolutely superior, compared with Sit + Unet_V as it can reduce more computation. As for the processing speed of a single image, Coformer+ SegCoformer is 258.26 ms, which is reduced by about 3 times compared to Sit + Unet_V. In the comprehensive analysis, the two-model combination method proposed in this study is suitable for efficient identification and fast grading of potato disease severity.

**Table 9.** Comprehensive performance comparison of the two combinations.

|  | CP/% | SP/% | KE/% | TSA/% | CT/ms | TFO/G |
|---|---|---|---|---|---|---|
| C + S | 99.06 | 89.85 | 1.46 | 84 | 771.46 | 128.08 |
| S + U | 97.86 | 88.50 | 2.04 | 83 | 258.26 | 4.03 |

## 4. Conclusions

This study proposed a two-stage approach for the rapid grading of potato disease severity and constructed experimental datasets for disease classification and semantic segmentation. The disease classification data comprised a total of 3000 images sourced from the publicly available datasets of PlantVillage and Kaggle. Moreover, the leaf semantic segmentation dataset was formed by selecting 1343 images from the disease classification dataset, focusing on early and late blight images. In addition, in order to improve the generalization ability of the model, different types of image enhancement were applied to the experimental data of disease classification and leaf semantic segmentation in this study.

Subsequently, this study combined the ConvNeXt V2 block with the Segformer block, adopting the design concept of the C-C-T-T combination. This design not only endowed the convolutional neural network with exceptional generalization capabilities but also harnessed the collective advantages of the self-attention mechanism's full field perception. Through this approach, a highly efficient and lightweight Transformer-based classification model named Coformer was developed. On the potato disease classification dataset, Coformer achieved an accuracy of 97.86%, all while maintaining a parameter count of merely 4.06 million. We added the MLP module for semantic information and image resolution reduction to Coformer, and developed a semantic segmentation model, SegCoformer, with an mIoU of 88.50% on the potato leaf semantic segmentation dataset, while the number of parameters was only 4.51 M, the method has a high accuracy while possessing a fast-processing speed, processing an image only takes 258.26 ms. These combined qualities enable its utilization in the swift grading of potato leaf severity.

Lastly, this study proposed a severity grading rule for potato diseases and employed a set of 100 potato leaf samples from the test dataset as subjects for image classification, image segmentation, and severity grading tests. The results indicated that the two model approaches introduced in this study effectively addressed the efficient identification and rapid grading of actual potato disease severity.

In summary, the method proposed in this study for the rapid severity grading of potato diseases meets practical application requirements, and the Coformer and SegCoformer models can also be applied to other datasets. However, during the research process, there still exist some potential areas for improvement and shortcomings. The adaptability and generalization performance of the models towards leaves at different growth stages is relatively insufficient. The background of the dataset is rather Singular, and there is a relatively limited amount of experimental data available for semantic segmentation. In future work, we intend to deploy the models on edge intelligent mobile devices for conducting field-based severity grading of potato diseases. This deployment will facilitate the processing of relevant decision-making information on lower-level devices. Additionally, we plan to collect more potato disease images from field conditions with complex backgrounds to use as training input for the models. This will enhance the models' generalization capabilities in complex background environments.

**Author Contributions:** Conceptualization, Y.X. and Z.G.; Methodology, Y.X. and Z.G.; Validation, Y.Z. and J.L.; Writing-Original Draft Preparation, Z.G.; Writing-Review & Editing, X.M., Z.G. and J.W.; Supervision, J.L. and Y.X.; Funding Acquisition, Y.X. All authors have read and agreed to the published version of the manuscript.

**Funding:** This research was funded by the Jilin Provincial Science and Technology Development Plan Project (20230202035NC) and the Jilin Provincial Science and Technology Development Plan Project (YDZJ202301ZYTS408).

**Institutional Review Board Statement:** Not applicable.

**Informed Consent Statement:** Not applicable.

**Data Availability Statement:** Dataset available on request from the authors.

**Conflicts of Interest:** The authors declare no conflicts of interest.

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
