# Peer review of "A Two-Stage Approach to the Study of Potato Disease Severity Classification"

_agriculture, doi:10.3390/agriculture14030386_

Round 1
Reviewer 1 Report
Comments and Suggestions for Authors#This manuscript, “A two-stage approach to the study of potato disease severity classification,” revealed that the rapid severity grading of both early blight and late blight in potato plants is innovative and addresses significant challenges in disease management. Here are some critical comments and considerations:
# line 44….to ……..In the Introduction, the statement, “The symptoms associated with both ailments manifest as scorching on tubers and leaves, presenting a formidable challenge in terms of differentiation and lending to widespread confusion,” should be supported by a few references……
# In the introduction and discussion, there is a strong need to add supportive references to justify your findings and comparatively show the importance of your findings. Authors may include a few references if they wish; otherwise from other they can take at least 10-15 references are required to support many statements in introduction and discussion;
#The detailed explanation of the Coformer and SegCoformer models is appreciated. However, more information on the architecture, training process, and hyperparameters could enhance the reproducibility and understanding of the proposed models.
#The description lacks information on the dataset used for training and testing. Details on data collection, size, diversity, and potential biases are crucial for assessing the generalizability of the proposed method.
# While the experimental results are promising, it would be beneficial to provide a more comprehensive comparison with existing methods. The Sit+Unet_V method is mentioned, but additional benchmarks and state-of-the-art approaches in the field should be considered for a more thorough evaluation.
# The study should address the generalization of the proposed models across different potato varieties, environmental conditions, and disease stages. Robustness testing under diverse scenarios will strengthen the reliability of the proposed method.
# The study should discuss the interpretability of the Coformer and SegCoformer models. Understanding how these models arrive at their decisions is crucial for building trust and facilitating their application in real-world scenarios.
# While classification accuracy and mean Intersection over Union (mIoU) are reported, additional metrics such as precision, recall, and F1-score could provide a more comprehensive evaluation of the models' performance, especially for imbalanced datasets.
# The study should explicitly mention any limitations of the proposed approach. Identifying potential challenges, such as sensitivity to certain conditions or types of noise in images, will guide future improvements.
# Considerations for the real-world implementation of the proposed method should be discussed. Factors such as computational requirements, scalability, and integration with existing agricultural systems are crucial for practical adoption.
# The study should briefly touch upon the ethical and social implications of using such technology in agriculture. This could include considerations related to data privacy, environmental impact, and the potential socio-economic effects on farmers.
# Discuss potential practical implications or applications of the findings for agricultural practices.
# Consider addressing any limitations or challenges encountered during the study.
# Include information on how the results were validated. Were the experiments repeated, and were similar results obtained?
#The conclusion should succinctly summarize the key findings of the study and reiterate the practical implications. Mention any limitations or avenues for future research.
Addressing these aspects would contribute to a more comprehensive and well-rounded research paper.
# In conclusion, while the proposed approach shows promise in addressing the challenges of disease severity grading in potato crops, further details, benchmarking, robustness testing, and consideration of broader implications will strengthen the overall contribution of the study.
Comments on the Quality of English LanguageMinor editing of English language required
Author Response
# line 44….to ……..In the Introduction, the statement, “The symptoms associated with both ailments manifest as scorching on tubers and leaves, presenting a formidable challenge in terms of differentiation and lending to widespread confusion,” should be supported by a few references……
Response: Accepted. We have added relevant references in Article 5
# In the introduction and discussion, there is a strong need to add supportive references to justify your findings and comparatively show the importance of your findings. Authors may include a few references if they wish; otherwise from other they can take at least 10-15 references are required to support many statements in introduction and discussion;
Response: Explained. We believe that we have already added enough references to demonstrate the importance of our research content. However, if you think there are any inappropriate or missing parts that need to be added, please provide suggestions and supplements.
#The detailed explanation of the Coformer and SegCoformer models is appreciated. However, more information on the architecture, training process, and hyperparameters could enhance the reproducibility and understanding of the proposed models.
Response: Explained. The model architecture has been elaborated in detail in Section 2.5, and information about the training process and hyperparameters has been detailed in Section 2.7. If you believe there are any omissions, please provide suggestions for additions.
#The description lacks information on the dataset used for training and testing. Details on data collection, size, diversity, and potential biases are crucial for assessing the generalizability of the proposed method.
Response: Explained. We have provided detailed information about the data in Sections 2.1 and 2.2. If you believe there are any omissions , please provide additions.
# While the experimental results are promising, it would be beneficial to provide a more comprehensive comparison with existing methods. The Sit+Unet_V method is mentioned, but additional benchmarks and state-of-the-art approaches in the field should be considered for a more thorough evaluation.
Response: Explained. Our experimental comparisons involve novel methods. For the classification task, we compared Vit, Sit, CoAt, MVit, and Coformer. For the segmentation task, we compared Deep_X, Deep_M, Unet_V, Unet_R, Segformer, and SegCoformer. We selected Sit+Unet_V for the final experimental comparison because it achieves the highest accuracy and mIoU in their respective tasks, as detailed in Section 3.5.
# The study should address the generalization of the proposed models across different potato varieties, environmental conditions, and disease stages. Robustness testing under diverse scenarios will strengthen the reliability of the proposed method.
Response: Explained. In future work, We will focus on evaluating the model's generalization capability across different potato varieties, environmental conditions, and disease stages.
# The study should discuss the interpretability of the Coformer and SegCoformer models. Understanding how these models arrive at their decisions is crucial for building trust and facilitating their application in real-world scenarios.
Response: Explained. We have thoroughly analyzed the performance of the models in Chapter 3 and presented the actual prediction results of the models.
# While classification accuracy and mean Intersection over Union (mIoU) are reported, additional metrics such as precision, recall, and F1-score could provide a more comprehensive evaluation of the models' performance, especially for imbalanced datasets.
Response:Explained. Sections 3.1 and 3.2 of the manuscript have conducted comparative analyses of the accuracy, recall, and F1 score of different models.
# The study should explicitly mention any limitations of the proposed approach. Identifying potential challenges, such as sensitivity to certain conditions or types of noise in images, will guide future improvements.
Response:Accepted. We have outlined the limitations of the proposed method and highlighted them in yellow in the conclusion section.
# Considerations for the real-world implementation of the proposed method should be discussed. Factors such as computational requirements, scalability, and integration with existing agricultural systems are crucial for practical adoption.
Response:Explained.This study focuses on researching methods for grading the severity of potato diseases. In the future, we also plan to deploy this in edge computing devices for the systematic development of agricultural systems.
# The study should briefly touch upon the ethical and social implications of using such technology in agriculture. This could include considerations related to data privacy, environmental impact, and the potential socio-economic effects on farmers.
Response:Explained.This study focuses on researching methods for grading the severity of potato diseases.The results indicated that the two model approaches introduced in this study effectively addressed the efficient identification and rapid grading of actual potato disease severity.
# Discuss potential practical implications or applications of the findings for agricultural practices.
Response:Explained.We have already outlined the potential practical impact on agricultural practices in both the introduction and conclusion sections.
# Consider addressing any limitations or challenges encountered during the study.
Response:Explained. We have already outlined the limitations encountered during the research process, highlighting them in green in the conclusion section.
# Include information on how the results were validated. Were the experiments repeated, and were similar results obtained?
Response:Explained. All experiments were repeated at least twice, validated, and yielded consistent results.
#The conclusion should succinctly summarize the key findings of the study and reiterate the practical implications. Mention any limitations or avenues for future research.
Response:Accepted.We have simplified the conclusion section and provided explanations on limitations or directions for future research.
Reviewer 2 Report
Comments and Suggestions for Authors
This is an interesting paper on the application of deep learning to the detection of potato diseases, via the analysis of plant leaves. The description of the approach is well done. The results (accuracy) are good and are well discussed. This is an interesting and robust contribution in terms of applying emerging deep learning techniques to agricultural applications.
This paper presents work that proposes a novel two-stage approach for the rapid severity grading of both early blight and late blight in potato plants. To validate the approach, extensive testing was conducted on a sizable dataset, and the achieved performance results were compared against established technique. The proposed method in this study targets for the rapid severity grading of potato diseases and pretends to meet practical application requirements. Two models, Coformer and SegCoformer, are used.The presented approach, tool and experiments, are a good contribution to the field of crop disease severity classification. The presented experimental results are well explained. Material and methods are also well presented and explained.
The comparison of classification performance of Coformer network (described in section 2.5.1) compared to other Transformer networks, presented and discussed in 3.2, revealed the ability of this network to maintain minimal model complexity while achieving good performance. Also the comparative experiment on potato disease severity grading presented in 3.5 shows the advantages and novelty of the presented approach, in terms of inference speed and floating-point computation. Conclusions are, in my opinion, consistent with the content and arguments presented along the paper. References are appropriate.Although some are more than 5 years old, they are justified. Data, figures and tables are fair. However fig 6 should be improved to make easier the observation of charts, maybe a resolution issue. The quality of figures 6 and 7, in terms of resolution, could be improved.
Author Response
The comparison of classification performance of Coformer network (described in section 2.5.1) compared to other Transformer networks, presented and discussed in 3.2, revealed the ability of this network to maintain minimal model complexity while achieving good performance. Also the comparative experiment on potato disease severity grading presented in 3.5 shows the advantages and novelty of the presented approach, in terms ofinference speed andfloating-point computation. Conclusions are, in my opinion, consistent with the content and arguments presented along the paper. References are appropriate.Although some are more than 5 years old, they are justified. Data, figures and tables are fair. However fig 6 should be improved to make easier the observation of charts, maybe a resolution issue. The quality of figures 6 and 7, in terms of resolution, could be improved.
Response: Accepted.We have updated clearer versions of Figure 6 and 7.
Reviewer 3 Report
Comments and Suggestions for Authors
Summary,
1. The main idea of this paper is the two-state lightweight model of deep learning approach for classifying potato disease severity. The model size was 17.51 MB that very lightweight. For future work, it may be suitable for mobile phones.
2. The proposed model is named Coformer and deployed the dataset from PlantVillage and Kaggle as more than 3,000 images.
3. The accuracy of Coformer is achievable at 97.86% for the validation model compared with the previous works.
Comment,
1. The title name should be renamed as "A two-state deep learning approach based potato disease severity classification" if possible.
2. The abstract should be not above 200 words, please rewrite it for readability and compact sentences.
3. Introduction, if possible please separate into two sections such as "Background, and Related works"
Author Response
The title name should be renamed as "A two-state deep learning approach based potato disease severity classification" if possible.
Response: Explained. The research in this paper focuses on the severity classification of potato leaves based on a two-stage deep learning approach. Therefore, we believe the original title of the paper is appropriate.
The abstract should be not above 200 words, please rewrite it for readability and compact sentences.
Response: Accepted. We have already condensed the word count of the abstract.
Introduction, if possible please separate into two sections such as "Background, and Related works"
Response: Explained.We believe that splitting it into two parts may disrupt the overall coherence and fluency of the paragraph. Keeping the introduction as a unified whole can better convey the background, purpose, and significance of the study.
Round 2
Reviewer 1 Report
Comments and Suggestions for Authors
The authors have revised the manuscript as per my expectation and now it is accepted in its current state.
Author Response
Accepted.